# Information theory bounds on randomness-based phase transitions

Noa Feldman, Niv Davidson, Moshe Goldstein

*Raymond and Beverly Sackler School of Physics and Astronomy, Tel-Aviv University, Tel Aviv 6997801, Israel*

We introduce a new perspective on the connection between many-body physics and information theory. We study phase transitions in models with randomness, such as localization in disordered systems, or random quantum circuits with measurements. Utilizing information-based arguments regarding probability distribution differentiation, rigorous results for bounds on critical exponents in such phase transitions are obtained with minimal effort. This allows us to rigorously prove bounds which were previously only conjectured for dynamical critical exponents in localization transitions. In addition, we obtain new bounds on critical exponents in many-body Fock space localization transition and localization in Coulomb-interacting models. Somewhat surprisingly, our bounds are not obeyed by previous studies of these systems, indicating inconsistencies in previous results, which we discuss. Finally, we apply our method to measurement-induced phase transition in random quantum circuits, obtaining bounds transcending recent mapping to percolation problems.

## I. INTRODUCTION

Over the years different connections have been recognized between information theory and physics, going from the Maxwell demon paradox [1], through using (quantum) information measures to bound renormalization group flows [2], to the black hole information problem [3]. Information theory concepts, such as complexity or entanglement, are used in many-body physics as a marker or a characteristic of phases [4–7] and in the study of the classical representability of many-body states [8–12]. Can one go further and use pure information or algorithmic complexity considerations to deduce the behavior of physical problems? Here only scant few results exist (see, e.g., Ref. [15] for a recent example in the context of the AdS/CFT duality). In our work, we propose a new approach for using information-theoretical arguments to obtain physical results, bounding the behavior of randomness-based phase transitions — seemingly unrelated to classical or quantum information.

The principal idea is as follows: We examine quantum phase transitions which rely on randomness, e.g., localization transitions. A model is defined by a probability distribution $\mathcal{P}_W$, where $W$ is proportional to $\mathcal{P}$'s width. A phase transition occurs when $W$ reaches a critical value $W_c$. We then ask: What is the required system size for the system to "know" its phase? The answer may be based on the physical properties of the system, depending on the correlation length characterizing the phase. It may also be based on information theory, requiring that the number of samples out of $\mathcal{P}_W$ would be sufficient to statistically determine $W$. Comparing both viewpoints gives an information-theoretical bound on the critical exponents of the phase transition.

Our method generalizes the rigorous proof of the Harris bound [16] as obtained by Chayes et al. [17, 18]. The generalization allows to apply the method to a vast range of models and obtain new bounds, as we demonstrate below.

The rest of this paper is organized as follows: In Sec. II A, we present our method by applying it to a paradigmatic phase transition, the Anderson localization (AL) transition [19–24]. We rederive the Harris bound as discussed above [16–18], using a much more intuitive approach, which allows to extend its validity to any smooth dependence of the disorder $\mathcal{P}_W$ on $W$, thereby complementing the argument. We discuss the extension of the method to classical temperature-driven phase transitions, the original realm of Harris's work, in Sec. II B. In Sec. II C, we adjust the method to bound dynamical critical exponents, and demonstrate the extension on non-Anderson transitions in Weyl and related systems, obtaining rigorous bounds which were previously only conjectured.

We then move to apply our method to interacting localization models, pointing at inconsistencies in previous results in two separate cases: In Sec. III A we apply our method to many-body localization (MBL) phase transition [37–39], studied in the setting of Fock space (FS) localization. We find discrepancies between our information-theoretic bounds and numerical results in limited-sized systems. Such discrepancies were already observed in MBL in real space [40, 41], and we observe an additional one in FS. This may be relevant to the ongoing discussion regarding the nature of MBL in the thermodynamic limit [42–64]. In Sec. III B we apply our method to localization transitions with Coulomb interactions and bound the dynamical critical exponent proposed for such a model [83]. Our bound is not obeyed by previous theoretical estimates in Ref. [83], which calls for additional investigation.

In Sec. IV we study measurement-induced transitions in random quantum circuits [85–88]. Results for such phase transitions have previously been obtained rigorously for a specific setting, namely, 1+1 dimensional circuit with Haar-random unitaries, with zeroth Rényi entropy (Eq. (16)) as the order parameter. We obtain a generic bound for all circuit settings, which is obeyed by existing numerical data. We summarize our conclusions and offer future outlook in Sec. V.

## II. INTRODUCTION OF THE METHOD

### A. Warmup example: Anderson localization

First, we demonstrate our method by rederiving the Harris bound on Anderson localization transitions. A rigorous proof equivalent to the following is presented in Appendix A, and here we give an intuitive overview.

We study a noninteracting tight-binding model with Hamiltonian $H_0$ with added disorder, i.e., random potentials,

$$H = H_0 + \sum_i \epsilon_i c_i^\dagger c_i, \tag{1}$$

where $c_i$ are annihilation operators, and $\{\epsilon_i\}$ are independent and identically distributed (i.i.d) variables drawn from a probability $\mathcal{P}_W(\epsilon)$. $W$ is called the *disorder parameter*, typically defined to be proportional to the $\mathcal{P}$'s width. We focus on lattice systems in spatial dimension $d$.

When $d > 2$, the eigenstates of the system with energy $E$ undergo a phase transition at a critical disorder $W_c(E)$ [22]. When $W > W_c(E)$, the system is in a localized phase [24]: eigenstates with absolute values of energy $> |E|$ are localized around a small group of lattice sites, their amplitude decaying exponentially with the distance from these sites. This phase is characterized by a localization length $\xi_E$. When $W < W_c(E)$ the system enters a diffusive phase at energy $E$ [24, 25], in which particles with absolute values of energy $< |E|$ scatter accross the lattice with distribution similar to that of a classical random walker, with a correlation length $\xi_E^{\mathrm{diff}}$. Assuming that the phase transition is second order, we define the critical exponent $\nu$:

$$\xi_E, \xi_E^{\mathrm{diff}} \sim |W - W_c(E)|^{-\nu}, \tag{2}$$

which governs the behavior in both phases due to renormalization group arguments [24].

We now introduce our method. Consider the following hypothetical setting: A classical computer receives an input composed of a model as in Eq. (1) above. The input also contains the values of $E$, $W_c(E)$, $\nu$, and all the coefficients necessary in order to describe the behavior of the system near criticality, up to any desired accuracy. The source of this prior knowledge is unimportant (e.g., a preliminary numerical computation using any amount of resources).

In the setting above, the classical computer is given a task: The computer is presented with a blackbox, emiting values distributed by $\mathcal{P}_{W_c(1\pm\delta)}$, where $\delta$ is small and given. It is then required to determine the sign, $\pm$.

A possible strategy for solving the task is to determine the physical phase defined by $\mathcal{P}_{W_c(1\pm\delta)}$. The computer samples $N$ values from the blackbox, and uses them to define local potentials $\{\epsilon_i\}$. It then simulates the behavior of a particle in a $d$-dimensional lattice of size $N$ in the studied model with energy $E$ (see Appendix A for technical details). Based on the simulation, the phase of the system is determined, and in turn, the sign $\pm$. The required system size for determining the phase is $N \sim \left[\max(\xi_E, \xi_E^{\mathrm{diff}})\right]^d \sim \delta^{-\nu d}$. The strategy proposed above is illustrated in Fig. 1.

We denote by $N_{\mathrm{opt}}$ the optimal number of samples required for differentiating the two probability distributions $\mathcal{P}_{W_c(1\pm\delta)}$. The information-theoretical requirement on $N_{\mathrm{opt}}$ is

$$\lim_{\delta \to 0} N_{\mathrm{opt}} \propto \delta^{-2}. \tag{3}$$

Eq. (3) can be understood intuitively, following Chebyshev's inequality. A more formal statement is provided in Appendix A.

It is required that $N_{\mathrm{opt}} \leq N$. Therefore,

$$\delta^{-2} = O(\delta^{-\nu d}) \quad \Rightarrow \nu \geq \frac{2}{d}. \tag{4}$$

We see that a bound on the critical exponent was obtained based solely on information-theoretical arguments, with minimal use of physical assumptions. We stress that while the setting above might be challenging to obtain, it is technically possible, which is sufficient to obtain a proof on the bound. The argument is based only on comparing number of samples in both hypothetical cases, rather than protocol complexity. The rigorous claim we make is the following: If the phase transition occurs and is described by Eq. (2), then necessarily $\nu \geq 2/d$. Note that the method only deals with the limit $\delta \to 0$, which corresponds to the thermodynamic limit , $N \to \infty$. A summary of the approach is presented in Fig. 2.

We note that usually, numerically-studied systems are simulated with a uniform disorder distribution rather than a smooth distribution. In this case, the bound obtained by our method is $\nu \geq 1/d$ (see Appendix A). Nevertheless, smooth and uniform distributions should behave similarly, since the latter may be considered a limiting case of the former. Indeed, numerical results obey the tighter bound, $\nu \geq 2/d$. We also note that one could use other quantities to distinguish the two phases, e.g., the localization properties of the eigenstate closest in energy to $E$, see Appendix B. This would parallel the Fock space discussion in Sec. III A below.

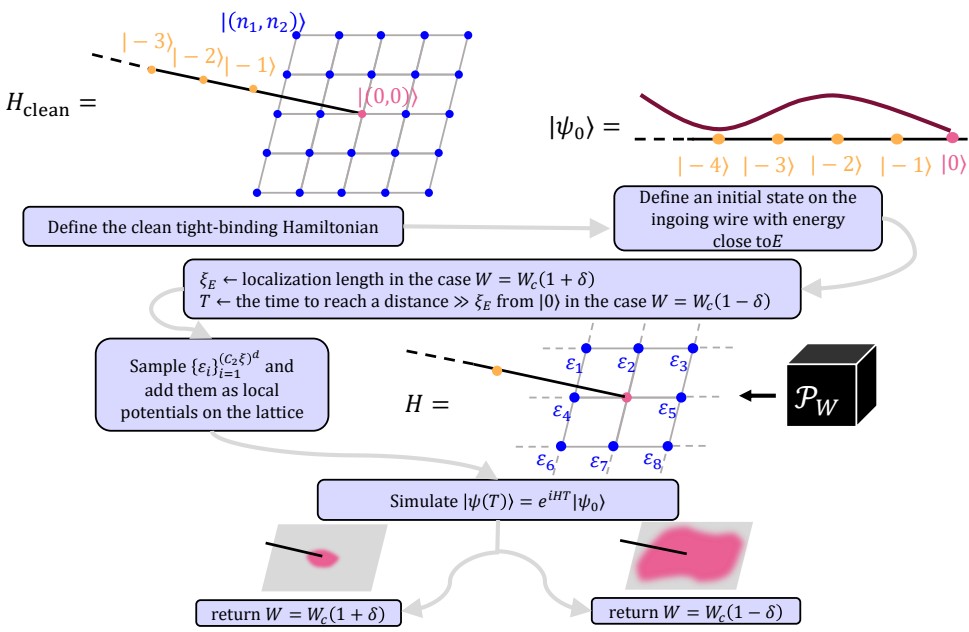

FIG. 1. Illustration of the protocol run (hypothetically) by the classical computer to distinguish the probability distributions $\mathcal{P}_{W_c(1\pm\delta)}$. The top sketch illustrates the simulated graph. A clean wire of length $L$ (orange nodes with negative indices) is coupled to the center (pink node, denoted by $|(0,0)\rangle$) of a $d$-dimensional lattice (blue nodes with positive indices $(n_1, n_2)$). An initial state of a free wave with energy $E$ is defined on the wire. Random potentials are added to the lattice sites with positive indices. The evolution of the state in time is simulated, and based on the amplitude of the state after a long time, the algorithm determines whether the system is in a localized ($W = W_c(1 + \delta)$) or delocalized ($W = W_c(1 - \delta)$) phase. Note that for clarity of the sketch, we use $d = 2$. However, the phase transition only occurs at $d > 2$.

## B. Harris bound for classical thermal phase transitions

One may also extend our argument to the systems originally considered by Harris [16], namely the classical ferromagnetic Ising model in $d$ space dimensions, with a small random addition $\Delta J$ to the exchange coupling $J$. Let us briefly recall Harris' argument. If the system is at a dimensionless temperature distance $\delta_T \equiv (T - T_c)/T_c$ from the clean critical temperature $T_c$, the correlation length is $\xi \sim |\delta_T|^{-\nu}$ in the clean case. Spins in regions of size $\sim \xi^d \sim |\delta_T|^{-\nu d}$ are thus correlated. The average disorder in such a region will be of order $\xi^{-d/2}$, leading to a correspondingly large shift in $\delta_T$. This shift is small with respect to the staring value of $\delta_T$, making the criticality robust to small disorder, only if $\nu \geq 2/d$.

We may rederive this result using our approach: Making the realistic assumption that small disorder shifts $T_c$ with some smooth dependence on the disorder distribution width, one may assume that near criticality, $T_c(1 \pm \delta_T)$ correspond, respectively, to disorder distribution width $W(1 \pm \delta)$ for some $\delta \sim \delta_T$. A decision problem may be defined to distinguish two probability distributions of widths $W(1 \pm \delta)$. A classical computer may then sample $N \sim \delta^{-\nu d} \sim \delta_T^{-\nu d}$ values from the distribution and determine the phase it corresponds to by calculating the partition function to sufficient accuracy, either by brute force or by, e.g., a Monte Carlo simulation running for a long enough time (recall that the only thing that matters to our argument is the number of disorder samples, not the calculation time). From the phase, the computer determines the distribution width and solves the decision problem. Comparing with $N_{\mathrm{opt}} \sim \delta^{-2}$, the Harris bound $\nu \geq 2/d$ is recovered.

## C. Dynamical critical exponents

Our approach may also be used to bound dynamical critical exponents, i.e., critical exponents that govern the behavior as a function of energy. We demonstrate it on the non-Anderson disorder-driven transitions in Weyl-like systems [30].

We focus on noninteracting Hamiltonians with short-range correlated random potential, such as the ones studied in Refs. [27–30]. It was shown there that a disorder-driven phase transition occurs in such systems even if localization is forbidden by topological or symmetry constraints. The transition is then manifested, e.g., by a change in the density

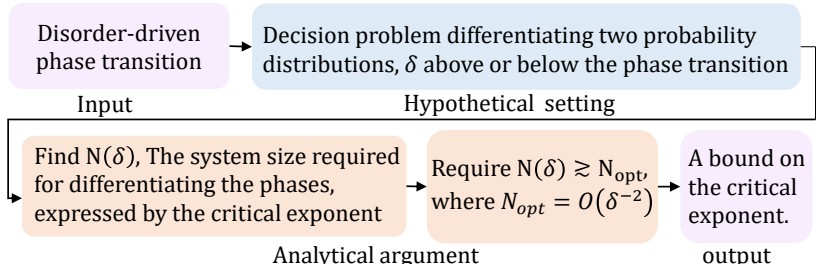

FIG. 2. A brief summary of the approach. We start with a disorder-driven phase transition assumed to be second order. A hypothetical decision problem is then considered, distinguishing between two probability distributions corresponding to two sides of the transition with a distance $\delta$ from the critical point. The $\delta$-dependence of the required system size in order to solve the problem is obtained in two ways: from physical considerations, based on the localization/correlation length, and from information theory, based on Eq. (3). The two requirements are compared to obtain a bound on the critical exponent.

of states (DoS) — the DoS at low energies vanishes when $W < W_c$, and becomes finite at larger disorder. The correlation length near the phase transition is governed by a critical exponent $\xi \sim |W - W_c|^{-\nu}$. The Harris bound on $\nu$ may be obtained similarly to the bound on the Anderson transition above.

The phase transition may also be approached by tuning the energy $E$ instead of the disorder, around a critical value $E_c$, which is fixed to the Weyl or Dirac nodes in the corresponding systems. Tuning the energy, we define the dynamical critical exponent $z$ as

$$\xi \sim |E - E_c|^{-z} . \tag{5}$$

Near $W_c$ and approaching $E_c$, the DoS $\rho$ is expected to behave as

$$\rho \sim |E - E_c|^{\frac{d}{z} - 1} . \tag{6}$$

We adjust our approach to bound $1/z$: Consider a disordered model with a critical energy $E_c$. One may present a hypothetical decision problem differentiating between two probability distributions with the same width, $W_c$, but different means: $E[\mathcal{P}_0] = 0, E[\mathcal{P}_{-\delta}] = -\delta$. Again, the required number of samples from $\mathcal{P}$ for differentiating the cases is $N_{\mathrm{opt}} = O(\delta^{-2})$.

A classical computer may sample $L^d$ values from the probability distribution and use it to define a disordered Hamiltonian. The computer may then calculate, e.g., the finite-size energy gap $\Delta$ in the spectrum around $E_c$. If the probability distribution is $\mathcal{P}_0$ the critical value is $E_c$, whereas for $\mathcal{P}_{-\delta}$ the spectrum shifts to lower energies and therefore the probed energy has a finite DoS. The expected behavior of $\Delta$ in the two phases is

$$\Delta \sim \begin{cases} L^{-z} & E = E_c, \\ L^{-d} \rho^{-1}(\delta) & E = E_c(1 + \delta). \end{cases} \tag{7}$$

The first (critical) case in Eq. (7) is derived from the definition of $z$, while the second is consistent with a finite DoS. The required number of samples, $N$, may be extracted by comparing the cases above, obtaining a requirement on the minimal $L$ needed to distinguish between the phases. By substituting $N = L^d$ and $\rho$ from Eq. (6), we obtain:

$$L^{-z} \propto L^{-d} \rho^{-1}(\delta) \quad \Rightarrow \quad N \propto \delta^{-\frac{d}{z}}, \tag{8}$$

and from the requirement $N \geq N_{\mathrm{opt}}$, the Harris bound

$$1/z \geq 2/d \tag{9}$$

is obtained, which was conjectured and empirically found to hold in Ref. 29.

Considering the numerical results obtained for $z$ [31–36], an interesting point arises: Unlike the real-space critical exponent $\nu$, $z$ tends to saturate (or come very close to saturating) the bound in Eq. (9). This implies that the (hypothetical) protocol above is equivalent to the optimal way of differentiating distributions. Put differently, it means that no information on $E[\mathcal{P}]$ is "thrown out" when computing the DoS or the energy gap $\Delta$. It would be interesting to characterize what makes this phase transition information-efficient.

## III.  LOCALIZATION TRANSITIONS IN INTERACTING MODELS

### A.  Fock space localization

We turn to obtaining new results on Fock space (FS) localization transition. The bound is not obeyed by previous numerical results [78], and we discuss this discrepancy below.

Adding interaction to Anderson-localized single-particle models may reintroduce ergodicity to the system. However, for a given interaction strength, it is widely believed that there is still a critical disorder value above which the system is in a many-body localized (MBL) phase, at least in 1D [37–39]. The interacting nature of MBL, combined with the fact that it occurs at excited states, limits heavily the accessible system size for numerical simulations of MBL. The study of MBL is thus required to navigate plausible theoretical arguments suitable for infinite-sized systems, and numerical results, limited to small-sized systems. This gap leads to challenges in the understanding of MBL and fuels a debate about whether such a phase even exists  [42–49, 56–64]. We note that another such debate arose about whether the Harris bound is obeyed by the real space MBL length [40, 41], and that our approach lends a more rigorous support to its applicability.

A recent enlightening angle on MBL originates from studying it in FS: the system can be thought of as a single particle scattered across the graph defined by the disordered Hamiltonian in FS. The study of MBL as FS localization allows to use single-particle techniques, and may shed light on MBL behavior [65–69]. In FS, the Hamiltonian can be written as a disordered tight-binding Hamiltonian with correlated local potentials: the number of sites in FS is $\mathcal{N} = d_{\mathrm{single}}^N$ where $d_{\mathrm{single}}$ is the Hilbert space size of a single site, but their random local potentials depend only on the $N$ parameters in real space.

A FS localization measure for an eigenstate $|\psi\rangle$ may be defined by [70–75]:

$$S_q \left( |\psi\rangle \right) = \frac{1}{1-q} \ln \left( \sum_{\alpha=1}^{\mathcal{N}} |\langle \psi \, | \, \alpha \rangle|^{2q} \right). \tag{10}$$

Using finite-size scaling, the behavior of $S_q$'s disorder average, $\overline{S_q}$, is predicted near the critical point  [74–76]:

$$\overline{S_q} \left( |\psi\rangle \right) = \begin{cases} \ln \mathcal{N} + b_{q,\mathrm{erg}} & \text{Ergodic (delocalized)} \\ D_q \ln \mathcal{N} + b_{q,\mathrm{MBL}} & \text{MBL} \end{cases}, \tag{11}$$

where $b_{q,\mathrm{erg}}$ is an emergent scale that goes to zero near the critical point, $D_q$ is a multifractal dimension, which goes to a critical value, $D_{q,c}$, at $W_c$, and $b_{q,\mathrm{MBL}}$ is an emergent scale which remains constant near the phase transition:

$$\begin{aligned} D_q - D_{q,c} &\sim \delta^{d\beta_q}, \\ b_{q,\mathrm{erg}} &\sim \delta^{-d\alpha_q}, \\ b_{q,\mathrm{MBL}} &\sim \mathrm{const.}, \end{aligned} \tag{12}$$

introducing two critical exponents, $\alpha_q, \beta_q$.

In Refs. 74, 77, a FS localization length is defined and assumed to scale with the same critical exponents as the real space localization length, in order to bound the FS behavior using the Harris criterion. We obtain an equivalent bound without relying on any such assumption. Near the critical point, $S_q$ varies continuously in the MBL phase while the ergodic phase displays a jump in $D_q$, as can be seen, e.g., in Ref.  75. We therefore focus on the requirement on the ergodic size to obtain a Harris-like bound is on the ergodic phase critical exponent $\alpha_q$:

$$\alpha_q \geq 2/d, \tag{13}$$

see Appendix B for details.

Interestingly, The FS transition has been studied in Ref. 78, for a one-dimensional 1/2-spin system with a uniformly distributed disorder and $q = 2$. The obtained critical exponent $\alpha_2$ was $\alpha_2 \approx 0.5$, which violates our rigorous bound (13). This implies either that the phase transition does not obey the expected form, or that the numerical results suffer from finite size effects. The discrepancy we uncover thus points at an overlooked mechanism, which may explain the inconsistency between the behavior in finite- vs. infinite- sized systems. The result above is in line with recent real-space numerical works which provide evidence that current numerically accessible system sizes might be too small [79–82], perhaps due to the avalanche mechanism [42–49, 56–64]; The latter implies Kosterlitz-Thouless scaling for the real-space correlation/localization length, corresponding to $\nu \to \infty$. This or another unaccounted-for mechanism might be necessary for a full physical picture of the FS behavior, which may result in a different behavior than the one in Eqs. (11), (12).

### B. Localization transition with Coulomb interaction

We apply our method to disordered models with long-range Coulomb interaction, uncovering an additional inconsistency in a previous result, as discussed below. The phenomenology of localization in this case is different than the standard MBL [83]. Particularly, it is found that in the delocalized regime (but not in the localized one) a dephasing length for excitations $L_\phi$ appears, which might be shorter than the correlation length $\xi$. Near and above the critical energy $E_c$, the dephasing length of an excitation at energy $E$ behaves as

$$L_\phi \sim (E - E_c)^{-\frac{1}{z}}, \tag{14}$$

where $z$ is the dynamical critical exponent. For $E < E_C$, in the localized regime, $L_\phi = \infty$, i.e., no inelastic decay. The localization length $\xi$ scales as $\xi(E)/\xi(E = 0) \sim (|E_C - |E||/E_C)^{-\nu}$, with the noninteracting exponent $\nu$. For $\nu z > 1$ one has $\xi > L_\phi$. Hence, the localized/delocalized phases are distinguishable already in systems of size $\sim L_\phi$.

We now bound the dynamical exponent: We introduce a hypothetical decision problem of differentiating two probability distributions with the same width and different means, $\mathbb{E}[\mathcal{P}_\pm] = \pm\delta/2$. The distributions may be distinguished by sampling $N \sim L_\phi^d$ values from the distribution and using them to define a system with Coulomb interaction and random disorder. If $\mathcal{P}_\pm = \mathcal{P}_+$, $E$ is in the delocalized regime and vice versa. The phase of the system may then be determined, e.g., by simulating the time evolution of an excitation of energy $E_c$. From this the sign of the critical energy shift, and therefore the probability distribution, could be identified. We now substitute Eq. (14) into the requirement $N \geq N_{\mathrm{opt}} \sim \delta^{-2}$ and obtain

$$1/z \geq 2/d, \tag{15}$$

provided that $\nu z > 1$, as explained above.

Intriguingly, the theoretical estimate provided for $z_\phi$ in Ref. 83 at $d = 3$ obeys the condition $\nu z > 1$ but violates Eq. (15). This may result from the inadequacy of using the Fermi golden rule to estimate the dephasing length in this regime [84].

## IV. MEASUREMENT-INDUCED PHASE TRANSITION IN RANDOM QUANTUM CIRCUITS

We proceed to obtain new bounds on measurement-induced phase transition in quantum circuits. Apart from the role played by randomness, these models are unrelated to localization, demonstrating the generality of our approach. Quantum circuits subject to random measurements [85–88] provide a generic model for open systems interacting with an environment, motivated both by quantum technology and many-body physics.

As a basic model, consider the following: The system is composed of a register of $L^d$ qubits, organized as a $d$-dimensional lattice of length $L$. $t$ layers of spatially local random 2-qubit unitary gates are then applied to the qubits, where in between each layer, each qubit is measured with probability $p$ in the $\{|0\rangle, |1\rangle\}$ basis. For $p$ close to 1, the qubits decay frequently into pure states, and the resulting state has little entanglement, typically an area-law. For small values of $p$, the system is close to not being measured at all, with a volume-law entanglement in the average case [85, 86]. A phase transition occurs at a critical value $p_c$.

The two phases are characterized by the entanglement between two parts of the system, denoted by $A$ and $\overline{A}$. A standard measure of entanglement is the von Neumann entropy, $\mathcal{S}(\rho_A) = -\mathrm{Tr}(\rho_A \log \rho_A)$, where $\rho_A = \mathrm{Tr}_{\overline{A}}(\rho)$ is the reduced density matrix of $A$. Due to the numerical and analytical inaccessibility of the von Neumann entropy, Rényi entropies are also introduced,

$$\mathcal{S}_n(\rho_A) = \log\left(\mathrm{Tr}(\rho_A^n)\right)/(1 - n), \tag{16}$$

which obey $\lim_{n \to 1} \mathcal{S}_n(\rho_A) = \mathcal{S}(\rho_A)$. The Rényi entropies are entanglement monotones, and for $n > 0$ they are continuous and reflect small changes in the entanglement accurately.

We follow Ref. 86 and define the characteristic length scale $\xi$ and time (circuit depth) scale $\tau$ by

$$\mathcal{S}_n(\rho_A(L, t, p)) - \mathcal{S}_n(\rho_A(L, t, p_c)) = f\left(\frac{L^d}{\xi^d}, \frac{t}{\tau}\right), \tag{17}$$

where $f$ is a scaling function. We then assume a second order phase transition:

$$\xi \sim \delta^{-\nu}, \quad \tau \sim \delta^{-z\nu}, \tag{18}$$

where $z = 1$ in the space-time symmetric case.

When $n = 0$, $d = 1$, and the unitary gates are Haar-random, the problem can be described as a graph percolation problem, each measurement becoming a cut in the circuit graph [86, 89]. This allows to leverage on the large body of knowledge on percolation problems [90]. We note in particular that a percolation phase transition must obey the Harris criterion.

The behaviors of Rényi entropies for other values of $n$ near the phase transition were studied numerically for $1 + 1$ dimensions [86], and they seem to obey the Harris criterion as well, although to the best of our knowledge, so far its validity had not been argued for theoretically. Note that the critical value $p_c$ extracted for $n \neq 0$ is different from the one extracted for $n = 0$ (which matched the percolation critical value, $p_c = 1/2$), suggesting that percolation might indeed not be a good model for the behavior of general Rényi entropies.

We now bound the critical exponents defined in Eqs. (18). The (hypothetical) decision problem may be differentiating two probability distributions over $\{0, 1\}$, where $\mathcal{P}(1) = p_c(1 \pm \delta)$. In order to differentiate the distributions, a classical computer may simulate a random circuit of size $L^d$ and depth $t$, with $N = tL^d$ possible spontaneous measurements, and compute $\mathcal{S}$ or $\mathcal{S}_n$. It is required that $L \gg \xi$ and $t \gg \tau$, i.e., $N = tL^d \gg \tau\xi^d \sim \delta^{-\nu(d+z)}$. The requirement $N \geq N_{\text{opt}}$ leads to

$$\nu(d + z) \geq 2. \tag{19}$$

The obtained bound is obeyed numerically [86, 91–93]. As opposed to existing analytical results, it is not limited to $d = 1$, $n = 0$, or Haar-random unitaries, but is completely general.

## V. CONCLUSION AND FUTURE OUTLOOK

We introduced a rigorous approach for obtaining critical exponent bounds in randomness-driven phase transitions, combining information bounds with physical phenomena. Due to its generality and the minimal amount of physical assumptions it requires, the bounds obtained by the approach are robust. By applying the method to several localization transitions, as well as measurement-induced phase transitions, we were able to obtain surprizing results in some models, and rigorous support to the previous conjectures in others.

Our approach may be applied to additional phase transitions, such as ones driven by correlated disorder or interdependent networks [96–102]. It can be readily applied to models with random hopping terms [17], spin glass models [103–105], localization critical exponents in Bethe lattices [106], and other aspects of the measurement-induced transition, such as the purification transition [91, 94] or the learnability transition [95]. Hopefully, the minimal effort required for obtaining such bounds, along with their robustness and potential for meaningful results, would encourage the wide use of our approach in further studies.

## ACKNOWLEDGMENTS

We thank D. Aharonov, A. Altland, I. Burmistrov, F. Evers, N. Laflorencie, A. Mirlin, and S. Syzranov for very useful discussions. Our work has been supported by the Israel Science Foundation (ISF) and the Directorate for Defense Research and Development (DDR&D) grant No. 3427/21 and by the US-Israel Binational Science Foundation (BSF) Grant No. 2020072. NF is supported by the Azrieli Foundation Fellows program.

## Appendix A: Re-obtaining the Harris bound for Anderson localization — Rigorous proof

In this section, we provide the full details of the proof made in Sec. II A in the main text, concentrating on the Anderson localization transition as a simple but paradigmatic example. We review in more details the physics of the system in Sec. A 1, then proceed to our argument in Sec. A 2. For the readability of this Appendix, some of the equations in the main text are repeated here.

### 1. The Anderson localization transition and the Harris bound

We start by introducing the model we focus on, which is a tight-binding model on a lattice with local disorder:

$$H = \sum_{\langle i,j \rangle} \left[ c_i^\dagger c_j + h.c \right] + \sum_i \epsilon_i c_i^\dagger c_i, \tag{A1}$$

where $\{\langle i,j\rangle\}$ are pairs of nearest-neighbor sites on the graph and $\{\epsilon_i\}$ are local random potentials. $\{\epsilon_i\}$ are independent and identically distributed (i.i.d) variables drawn from a probability distribution $\mathcal{P}_W(\epsilon)$, where $\langle\epsilon\rangle_{\mathcal{P}_W}$ can be taken as 0. $W$ is called the *disorder parameter* or *strength*, and is typically defined to be proportional to the standard deviation of $\mathcal{P}$. The spatial dimension of the lattice is denoted by $d$.

By scaling arguments, Ref. [22] shows that when $d > 2$, the eigenstates of the system with energy $E$ undergo a localized-diffusive phase transition at a critical disorder parameter $W_c(E)$. When $W > W_c(E)$, the system is in a localized phase [24], and eigenstates with absolute values of energy $> |E|$ are localized around a small region in the lattice with an exponentially-decaying amplitude with the distance from this region. This results in a diffusive behavior only over a small localized area of the lattice, its volume of order $\xi_E^d$, where $\xi_E$ is called the *localization length*. Assuming that the phase transition is second order, we define the critical exponent $\nu$ of the localization phase transition:

$$\xi_E \sim \delta^{-\nu}, \tag{A2}$$

where $\delta = W - W_c(E)$, characterizing the phase transition as it is approached from above $W_c(E)$.

When $W < W_c(E)$ the system enters a diffusive phase at energy $E$ [24, 25]: Eigenstate with absolute value of energy $< |E|$ divide the lattice into 'boxes' of size $\left[\xi_E^{\mathrm{diff}}\right]^d$, each appearing locally as if the state is localized at the center of the box, with amplitudes overlapping at the boxes' edges. Due to this overlap, particles scattered across the lattice with energy expectation value $< |E|$ in absolute value (assuming a sufficiently narrow energy spread) may diffuse from one box to another across the entire lattice, with distribution similar to that of a classical random walker. The probability distribution on the graph for such a process, starting at site $i$, is, to a first approximation, the Rayleigh distribution,

$$\mathbb{E}_{\mathcal{P}_W}\left[\left|\langle i|\,e^{-iHt}\,|j\rangle\right|^2\right] \propto \frac{1}{\left(D\left(E\right)t\right)^{d/2}}e^{-\frac{|\vec{r}_j-\vec{r}_i|^2}{4D(E)t}}, \tag{A3}$$

where $D(E) \sim \left[\xi_E^{\mathrm{diff}}\right]^{-(d-2)}$ is called the diffusion coefficient, which is proportional to the conductivity, and $\vec{r}_i, \vec{r}_j$ are the positions of sites $i, j$, respectively. Approaching the phase transition from within the diffusive phase, the phase transition is again characterized by the same critical exponent $\nu$ due to renormalization group arguments [24],

$$\xi_E^{\mathrm{diff}} \sim \delta^{-\nu}. \tag{A4}$$

The original argument leading to the Harris criterion [16] does not apply for the Anderson transition, but the criterion itself, namely

$$\nu \geq \frac{2}{d}, \tag{A5}$$

can be shown to hold using the mathematically rigorous approach of Refs. [17, 18]. The latter does not, however, lend itself straightforwardly to generalizations, as opposed to the information-theoretical view we present below.

We note that in the case that the critical exponents are not equal on both sides of the phase transition (i.e., the critical exponents in Eqs. (A2, A4) are different), the original bound, as well as the bound we obtain below, applies for the larger critical exponent among the two.

## 2. Information-theoretical bound on the critical exponent

We consider a model undergoing an Anderson localization phase transition. Near criticality, the required system size in order to distinguish the localized and delocalized phases is set by $\xi$. This size is then compared with the minimal number of samples from the disorder distribution, to obtain a bound on $\nu$.

The rigorous argument goes as follows. As a thought experiment, consider the following setting: We are given a model of the form of Eq. (A1) on a lattice of dimension $d > 2$. Suppose that we are also given the values of some energy $E$, the critical disorder $W_c(E)$, and the critical exponent $\nu$, all up to any desired accuracy. $\mathcal{P}_W(\epsilon)$ is defined by $W = W_c(E)\left(1 \pm \delta\right)$, where $\delta$ is small and given, and the sign $\pm$ is unknown. Lastly, we know the diffusion constant $D(E)$ and correlation length $\xi_E^{\mathrm{diff}}$ for such a model for $W = W_c(1 - \delta)$, the localization length $\xi_E$ for $W = W_c(1 + \delta)$, and the coefficients necessary in order to turn Eqs. (A2),(A3) into equalities.

For some models, $W_c(E)$ and $\nu$ have been found analytically [24]. However, in our approach the source of knowledge is unimportant (it can be obtained, for example, by a preliminary numerical computation which uses an arbitrarily large amount of resources). We stress that it is by no means required that this setting is actually implemented in real

life, as it may require much more prior knowledge than one might have in practice. It is rather sufficient that the setting is in principle possible to realize for our information-theoretical bound to hold, and it is from this requirement that the Harris bound is extracted.

In the hypothetical setting, a classical computer is given a decision problem: Given a sampling access to the distribution $\mathcal{P}_{W_c(1\pm\delta)}$, determine whether the sign $\pm$ is positive or negative. The task is, in fact, differentiating between two possible distributions, $\mathcal{P}_{W_c(1\pm\delta)}$.

Viewed from a different angle, the question is whether the system defined by $\mathcal{P}_W(\epsilon)$ is in the localized or delocalized phase. One may consider a solution protocol that relies on the fact that the two distributions lie on two sides of a phase transition, and utilizes the prior knowledge on the model: The computer simulates a $d$-dimensional lattice as in the studied model. An initial state is chosen to be arbitrarily close to an eigenstate with energy $E$, by adding a disorder-free wire (long enough so that its level spacing is smaller than that of the disordered system) entering the lattice and fixing the initial state to be close enough in energy to $E$ on this wire (see Fig. 1 in the main text). The computer then chooses constants $C, C' \gg 1, c \lesssim 1$, independent of $\delta$, and computes the time $T \sim \xi_E$ it would take a particle to reach a distance $C\xi_E$ from the entry point with probability $c$ in the diffusive case in a system of size $(C' \cdot C \cdot \xi)^d$.

Then, using $N = (C \cdot C' \cdot \xi_E)^d$ actual samples of $\mathcal{P}_W$ as the local potentials, the computer defines a disordered model. The computer simulates the time evolution of the initial state up to time $T$ in the disordered lattice. After the simulation, the probability of the particle to be at a distance $\geq C\xi_E$ from the initial site is calculated. If the probability is high enough, that is, above $c$, it can be deduced that the simulated system is in the diffusive phase and $\delta < 0$, and vice versa. The algorithm is sketched in Fig. 1 in the main text and presented rigorously in Table I.

Let us analyze the dependence of $N$ on $\delta$. The evolution is simulated only up to a finite time $T$. $T$ was chosen such that the number of sites on which the wavefunction amplitude become non-negligible during the evolution in the diffusive case is proportional to $\xi_E^d$, so only $\sim \xi_E^d$ sites, that is, $N \sim \xi_E^d$ samples, need to be considered.

We now analyze the information-theoretic requirement on $N$. $N_{\text{opt}}$ denotes the required sample size in order to solve the decision problem. It is expected that

$$\lim_{\delta \to 0} N_{\text{opt}} \propto \delta^{-2}. \tag{A6}$$

The above can be understood intuitively: The required task is to estimate $W$, which is proportional to $\left\langle \epsilon^2 \right\rangle_{\mathcal{P}_W}$, up to an estimation error proportional to $\delta$; Eq. (A6) follows from Chebyshev's inequality. It may also be obtained rigorously as follows: In the asymptotic case of $\delta \to 0$, it was shown [26]:

$$N \propto d_H^{-2} \left( \mathcal{P}_{W_c(1+\delta)}, \mathcal{P}_{W_c(1-\delta)} \right), \tag{A7}$$

where $d_H(p, q)$ is the Hellinger distance between the distributions $p, q$:

$$d_H(p, q) = \sqrt{\int \frac{1}{2} \left( \sqrt{p(x)} - \sqrt{q(x)} \right)^2 dx}. \tag{A8}$$

For a small enough $\delta$ and a smooth distribution $\mathcal{P}_W$, the Hellinger distance is linearly dependent on $\delta$ to first order, and Eq. (A6) is obtained.

Comparing with Eq. (A6), we obtain the result

$$\lim_{\delta \to 0} N \geq \lim_{\delta \to 0} N_{\text{opt}}$$
$$\Rightarrow \xi_E^d \sim \delta^{-\nu d} \geq \delta^{-2} \tag{A9}$$
$$\Rightarrow \nu \geq \frac{2}{d}, \tag{A10}$$

which completes the proof. Note again that the time and memory complexity required for the system simulation is ignored, as the result relies only on sample complexity.

The rigorous statement we obtain is the following: If the phase transition occurs, and is a second-order phase transition which obeys Eqs. (A2), (A4), then it is required that $\nu \geq 2/d$. If this requirement is not obeyed by some numerical or plausible theoretical argument, one may deduce either that the assumptions regarding the nature of the phase transitions were false, or that the results were inaccurate, due to, e.g., too small system sizes used numerically or the inadequacy of some plausible argument.

It is worthwhile to mention that for the most part, numerically-studied systems are simulated with a uniform distribution rather than a smooth distribution. A single sample in the range $\pm [W_c(1 - \delta), W_c(1 + \delta)]$ would be enough for differentiating the distributions, and therefore the number of required samples in this case is $N_{\text{opt}} \sim$

$\frac{W_c(1+\delta)}{2W_c\delta} = O\left(\delta^{-1}\right)$, as the Hellinger distance would indeed show. Thus, the bound obtained by our method would be $\nu \geq 1/d$. However, it seems that numerical results for the uniform distribution still obey the tighter bound, $\nu \geq 2/d$. This is not surprising, since a uniform distribution may be thought of as the limit of a smooth distribution, and the physical behavior is expected to be similar.

Finally, we note that the (hypothetical) protocol suggested above and in Table I may fail if, by some rare event, the local potentials drawn from $\mathcal{P}_{W(1\pm\delta)}$ results locally in the opposite phase from the one expected, that is, delocalization if $\delta > 0$ or vice versa. However, repeating the process would decrease the probability for it up to an arbitrarily small one, exponentially fast in the number of repetitions, such that the relation $N \sim \delta^{-\nu d}$ remains valid for arbitrarily small probability of failure.

Let us finally note that here, we used the scattering of a wavepacket as the test property for distinguishing between the phases. However, a variety of properties may be used instead, e.g., the localization properties of the eigenstate closest in energy to $E$. We make use of this in the derivation of the Fock space localization critical exponent in Appendix B below.

## Appendix B: Harris-like bound for Fock space localization transition

Here we bring the full details of the bound obtained for the critical governing the approach of the Fock space (FS) MBL from within the ergodic phase, discussed in Sec. III A in the main text. First, we review the model and the phase transition, as done in the main text, in Sec. B 1. We then derive the bound on the critical exponent in Sec. B 2, and finally discuss the discrepancy between our bound and the numerical results of Refs. [75, 78] in Sec. B 3.

### 1. Fock space localization transition

We consider a disordered model such as in Eq. A1, with added interactions between the particles. The interactions may destroy the localization that emerges in the noninteracting case, but it is still widely believed that for a high enough disorder, an ergodic-localized phase transition occurs.

We study this system in FS rather than real space. Here the state of the interacting system may be viewed as that of a single particle scattering across the FS. In FS, the Hamiltonian can be written as Eq. (A1) with correlated local potentials: The number of sites in FS is $\mathcal{N} = d_{\text{single}}^N$, where $d_{\text{single}}$ is the Hilbert space of a single site, but their random local potentials depend only on the $N$ parameters $\{\epsilon_i\}_{i=1}^N$. Due to the long-range correlations of the random potentials, the localized phase in real space behaves as a critical phase in FS.

In analogy to previous treatment of Anderson localization [70], and following Refs. [71–74], Ref. [75] introduces a FS localization measure for an eigenstate $|\psi\rangle$ of the system, brought in Eqs. (10), (11), and (12) in the main text. In the next section, we bound the behavior of the critical exponent $\alpha_q$ of Eq. (12), which governs the behavior in the ergodic phase, $W > W_c$.

### 2. Bounding the Fock space localization transition

As in the Anderson localization case, a classical computer may be hypothetically required to differentiate two probability distributions, $\mathcal{P}_{W_c(1\pm\delta)}(\epsilon)$ (again, in a thought experiment in which all prior knowledge of the model is accessible, disregarding the space and time computational costs). The problem may be solved by performing $N$ samples from the distribution $\mathcal{P}$ and using them to define an $N$-site interactive disordered Hamiltonian. The Hamiltonian is then diagonalized and $S_q$ is computed. The cases $W_c(1\pm\delta)$ are distinguished based on the proximity of $S_q$ extracted from the simulated system and its expected value for both cases as in Eq. (11). The protocol is presented more rigorously in Table II.

We analyze the requirements for the success of the protocol presented in Table II. $\Delta S_q$ is defined to be the deviation of $S_q$ from $\overline{S_q}$. The protocol is successful if

$$\Delta S_q \ll \left| \overline{S_{q,\text{erg}}} - \overline{S_{q,\text{MBL}}} \right|. \tag{B1}$$

| Initialization($E, d, \delta$): | |
|---|---|
| $0 \leftarrow$ starting_site<br>$c \leftarrow$ small constant $\lesssim 1$<br>$C \leftarrow$ large constant $\gg 1$<br>$C' \leftarrow$ large constant $\gg 1$<br>$\varepsilon \leftarrow$ small constant $\ll C^{-d}$ | Choose ancillary constants. |
| $k \leftarrow \cos^{-1} \frac{E}{2}$<br>$L \leftarrow \left\lceil \left(\delta^{\nu d} \varepsilon\right)^{-1} \right\rceil$<br>$|\psi_0\rangle \leftarrow \frac{1}{\sqrt{n}} \sum_{j=-L}^{0} \sin(kj) |j\rangle$<br>$H_{\text{wire}} \leftarrow \sum_{j=-L}^{0} \left(|j\rangle \langle j+1| + \text{h.c}\right)$<br>$n \leftarrow \sum_{j=-L}^{0} \sin^2(kj)$<br>$\xi_E \leftarrow$ localization_length $(E, \delta)$<br>$H_{\text{clean}} \leftarrow \sum_{\langle i,j \rangle i,j \in 1 \ldots (CC'\xi_E)^d} |i\rangle \langle j| + \text{h.c.}$<br>$H \leftarrow H_{\text{wire}} + H_{\text{clean}}$ | Construct the Hamiltonian of a clean $d$-dimensional lattice of size $(C'C\xi)^d$, and attach to it a clean wire which is long enough such that its level spacing near $E$ is smaller. The initial state is confined to the wire, with energy expectation value close to $E$. The system is illustrated in Fig. 1 in the main text. |
| $T \leftarrow \min \left\{ t \mid \mathbb{E}_{\mathcal{P}_{W_c(1-\delta)}} \sum_{j \text{ s.t. } |r_j - r_0| \geq \lceil C\xi_E \rceil} \left[ \left| \left\langle \psi_0 \left| e^{i(H + \sum_{i=1}^{(CC'\xi)^d} \epsilon_i |i\rangle \langle i|)t} \right| j \right\rangle \right|^2 \right] \geq c \right\}$ | Choose a time $T$ long enough for a diffusive particle on a $d$-dimensional lattice to reach a distance of $C\xi_E$ from the 0 site with probability $c$, assuming the system is diffusive (that is, $W = W_c(1-\delta)$). |
| $P_{\text{exp},d} = \mathbb{E}_{\mathcal{P}_{W_c(1-\delta)}} \left[ \sum_{j \text{ s.t. } |r_j - r_0| \geq \lceil C\xi_E \rceil} \left| \left\langle \psi_0 \left| e^{i\left(H + \sum_{i=1}^{(CC'\xi)^d} \epsilon_i |i\rangle \langle i|\right)T} \right| j \right\rangle \right|^2 \right]$<br><br>$P_{\text{exp},l} = \mathbb{E}_{\mathcal{P}_{W_c(1+\delta)}} \left[ \sum_{j \text{ s.t. } |r_j - r_0| \geq \lceil C\xi_E \rceil} \left| \left\langle \psi_0 \left| e^{i\left(H + \sum_{i=1}^{(CC'\xi)^d} \epsilon_i |i\rangle \langle i|\right)T} \right| j \right\rangle \right|^2 \right]$ | Based on Eq. (A3), find the expected probability that a particle reached a distance $C\xi_E$ in the localized and delocalized phases, respectively. |
| **return** $C, H, T, P_{\text{exp},d}, P_{\text{exp},l}, \xi_E$ | |

| determine_delta_sign($E$, $d$, $\delta$, $\mathcal{P}$): | |
|---|---|
| $C, H, T, P_{\text{exp},d}, P_{\text{exp},l}, \xi_E \leftarrow$ **Initialization**($E, d, \delta$) | |
| **for** $i$ **in** $\left\{ 1 \ldots (CC'\xi)^d \right\}$ **do**<br>  $\epsilon_i \leftarrow$ **Sample** $(\mathcal{P})$<br>  $H \leftarrow H + \epsilon_i |i\rangle \langle i|$ | Construct a Hamiltonian of a subsystem of volume $(C')^d = O\left(\delta^{-\nu d}\right)$. Based on Eq. (A3) we see that the amplitude on sites $i$ for which $|r_i - r_0| \geq \sqrt{D(E)T} \sim \delta^{-\nu d}$ decays exponentially, and therefore considering a graph of this size will result in a good separation of the phases. |
| $|\psi(T)\rangle = \exp(-iHT) |\psi_0\rangle$<br>$P_{C\xi_E} = \sum_{j \text{ s.t. } |r_j - r_0| \geq \lceil C\xi_E \rceil} |\langle \psi(T) | j \rangle|^2$ | Calculate the state $e^{-iHT} |\psi_0\rangle$, and from it deduce the probability for a particle to be at a distance larger than $\lceil C\xi \rceil$. |
| **if** $|P_{C\xi_E} - P_{\text{exp},d}| > |P_{C\xi_E} - P_{\text{exp},l}|$ **do**<br>  **return** $\delta < 0$.<br>**else do**<br>  **return** $\delta > 0$. | If the particle has reached a distance of $\lceil C\xi \rceil$ or more with high probability, we conclude that the simulated system in the diffusive phase and $\delta < 0$. Otherwise, we conclude that $\delta > 0$. |

TABLE I. The suggested algorithm that utilizes the AL transition in order to differentiate the probability distributions. The algorithm is presented on the left column, and on the right we provide explanations.

Dividing by $\ln \mathcal{N}$ and substituting Eq. (12), we obtain the requirement

$$\Delta S_q / \ln \mathcal{N} \ll \left| \overline{S_{q,\text{erg}}} - \overline{S_{q,\text{MBL}}} \right| / \ln \mathcal{N}$$
$$= \left| 1 - D_q(\delta) + \frac{b_{q,\text{erg}}(\delta)}{\ln \mathcal{N}} - \frac{b_{q,\text{MBL}}(\delta)}{\ln \mathcal{N}} \right|. \tag{B2}$$

The behavior presented in Eqs. (12),(11) is apparent for large system sizes, that is, $\ln \mathcal{N} / b_{q,\text{erg}} \gg 1, \ln \mathcal{N} / b_{q,\text{MBL}} \gg 1$. Approaching the critical point from above (localized phase), $S_q$ varies continuously to its critical value $S_{q,c}$, while the ergodic (critical) phase one expects a jump in $D_q$ as $W$ is varied towards $W_c$, see, e.g., in Ref. [75]. The above

implies that it is the ergodic phase which defines the size of the system in which the phases would be distinguishable. This leads to the requirement

$$\ln \mathcal{N} \gg b_{q,\mathrm{erg}}. \tag{B3}$$

In addition, we need $\Delta S_q$ to obey Eq. (B2). Since the right hand side of Eq. (B2) is constant in the thermodynamic limit, this requirement is met provided $\Delta S_q / \ln \mathcal{N}$ decreases with increasing system size. This is found to be the case numerically [78], as we further discuss below. Following Eqs. (12),(B3), the former requirement may be translated to $N \sim \delta^{-d\alpha_q}$. The requirement $N \geq N_{\mathrm{opt}}$ results in a Harris bound on $\alpha_q$,

$$\alpha_q \geq 2/d. \tag{B4}$$

As mentioned above, this bound was already presented in Ref. [74], relying on the assumption that the real space and Fock space phase transition displays the same behavior. Here we obtained the bound without such an assumption.

### 3. Fock space localization bound: Inconsistency with numerical results.

In the previous section, a Harris-like bound was obtained on the critical exponent $\alpha_q$ all all values of $q$. The FS localization transition has been studied in Refs. [75, 78], for a one-dimensional system with a uniformly distributed disorder and $q = 2$. The obtained critical exponent $\alpha_2$ was $\alpha_2 \approx 0.5$, and the fluctuations $\Delta S_2 / \ln \mathcal{N} \sim N^{-1/2}$. This implies that $N = \delta^{-\mu}$ where $\alpha_2 \leq \mu \leq 1$ may be sufficient to obey Eq. (B2) and differentiate the distributions, which contradicts our argument: it implies that simulating a many-body system with disorder and determining its phase is a better strategy for differentiating probability distributions than a standard statistical test. Note that even if the behavior of the uniform disorder distribution is different than that of smooth distributions, the requirement would be $\alpha_2 \geq 1$. The numerical results remain inconsistent with our bound even in this relaxed case. We put our result form the previous section rigorously: If the Fock space transition occurs, it is described by Eqs. (11), (12), and $\Delta S_q / \ln \mathcal{N}$ is small for large enough $N$, then it follows that $\alpha_q \geq 2/d$ for all $q$.

| **Initialization**$(E, \delta, \mu)$**:** | |
|---|---|
| $0 \leftarrow$ starting_site $c \leftarrow$ const., $0 < c < 1$ $N \leftarrow \lceil \delta^{-d\alpha_q c} \rceil$ $\mathcal{N} \leftarrow d_{\mathrm{single}}^N$ $H \leftarrow H_{\mathrm{clean}}(N)$ $q \leftarrow$ const. $> 1$ | Choose ancillary coefficients. Choose a system size $N = \delta^{-d\alpha_q c}$, where $c$ is a positive constant smaller than 1, which guarantees that in the vicinity of the phase transition, $\frac{b_{q,\mathrm{erg}}}{N} \to 0$. Start with the $N$-particle disorder-free Hamiltonian. $d_{\mathrm{single}}$ denotes the Hilbert space size of a single site. Choose $q > 1$ for the FS-locality measure, $S_q$. |
| **determine_delta_sign():** | |
| **for** $i$ in $1..N$ **do** $\quad h_i \leftarrow$ **Sample**$(\mathcal{P})$ $\quad H \leftarrow H + h_i \sigma_i^z$ | Add the disorder to the Hamiltonian by performing $N$ samples from the distribution. |
| $\lvert \psi_E \rangle \leftarrow$ **eigenvector**$(H, E)$ $S_q \leftarrow -\ln \left( \sum_{\alpha=1}^{\mathcal{N}} \lvert \langle \psi_E \mid \alpha \rangle \rvert^{2q} \right)$ | Diagonalize $H$ to obtain the eigenvector with eigenvalue closest to $E$ and calculate its $S_q$. |
| **If** $\left\lvert \frac{S_q}{\ln \mathcal{N}} - \left( 1 + \frac{b_{q,\mathrm{erg}}(\delta)}{\ln \mathcal{N}} \right) \right\rvert < \left\lvert \frac{S_q}{\ln \mathcal{N}} - \left( D_q(\delta) + \frac{b_{q,\mathrm{MBL}}(\delta)}{\ln \mathcal{N}} \right) \right\rvert$ **do** $\quad$ **return** $\delta < 0$. **else do** $\quad$ **return** $\delta > 0$. | Determine the phase of the simulated system based on the calculated $S_q$. |

TABLE II. The suggested algorithm that utilizes the FS localization transition in order to differentiate the probability distributions. The algorithm is presented on the left column, and on the right we provide explanations.

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
