# Peer review of "Information theory bounds on randomness-based phase transitions"

_SciPost Physics_

## Round 1 · Referee Report · Anonymous (Referee 1) · 2025-1-28

Report

Dear Editor,

The work by Noa Feldman and collaborators discusses information-theory-based bounds on critical exponents for disordered systems. Overall, the arguments presented are concise and clear: the method elegantly derives Harris bounds and generalizations to dynamical settings, supporting conjectured hyperscaling relations for the dynamical critical exponent.

This paper also raises significant challenges for the field of many-body localization (MBL). The authors highlight an inconsistency between their analytically motivated arguments and numerical finite-size scaling (FSS) results for Hilbert space delocalization, as measured by the participation entropy in the many-body computational basis. Indeed, the numerical results appear to violate the authors’ information-theoretic bounds, which could serve as a sine qua non criterion for the validity of FSS in ergodicity-breaking problems.

The authors also discuss the case of measurement-induced phase transitions (MIPTs) in random circuits, employing arguments based on entanglement Rényi entropies. I would encourage the authors to extend their analysis to the case of Hilbert space delocalization (see https://doi.org/10.1103/PhysRevLett.128.130605), where significantly larger system sizes are accessible for Clifford circuits. This setting, distinct from MBL, features a transition driven by the subleading coefficient $c_q$ rather than the (multi)fractal dimension $ D_q$. Including this analysis would further strengthen the authors’ results and align the presentation of their findings more cohesively.

A noteworthy figure of merit not addressed in the manuscript is that the argument extends to noisy (and open) dynamical systems (see https://doi.org/10.1103/PhysRevX.11.031066, https://doi.org/10.1103/PhysRevLett.132.140401, https://doi.org/10.1103/PRXQuantum.5.030327). In such cases, where $z \to \infty$ , the bound on $\nu$ remains valid.

Overall, with these minor inclusions, I strongly support the publication of this paper in SciPost Physics.

Requested changes

see report

Recommendation

Publish (surpasses expectations and criteria for this Journal; among top 10%)

---

## Round 1 · Referee Report · Anonymous (Referee 2) · 2025-2-26

Strengths

1) Information-theoretic approach to Harris criterion 2) Succesfull application to a variety of one-particle Anderson models (known results)

Weaknesses

1) Apparently it does not apply to many-body localization 2) They did not try to apply it to the Anderson model on the regular random graph which is a toy model for the MBL transition, if it exists

Report

The paper revisits the well-known Harris bound using an information theoretic treatment.

The argument is not that different from the usual one: one must have a sufficiently large system/sample size to distinguish two probability distributions which are close to each other. The differentiation is interpreted in terms of "calls" of a classical computer which runs a dynamics. Although the dynamics is quantum, so one should really talk about a quantum computer.

Nonetheless, the authors discuss the Anderson model and the "Fock space localization," which is another name for the MBL phase and transition.
Here they find a discrepancy between their treatment and the numerics. This is also well known (at least since their ref. [87] was published). They suspect this is due to the fact that the MBL transition is not a vanilla second order phase transition. They might be right (this was also stated in many previous papers, see the discussion in [D]) but they fail to prove their statement.

One way to do it would be to study the one case in which the Anderson transition itself is not a simple second order phase transitions: this is the limit d\to\infty where it becomes Kosterlitz-Thouless like (see Refs. [A,C]). The authors do not seem to be aware of these papers which instead are central to understand the difference between localization in finite and infinite dimensions, while they cite other papers which have made similar claims but do not present a clear picture of the why a KT transition is present here.

If, instead of looking at the leading order term in S_q, like they do in their eq. (11), they looked at the derivative with respect to the logarithm of the system size, they would turn the problem into distinguishing different orbits of the fractal dimensions D_q which evolve as a function of \ln N. Some orbits end on the localized region, some will bounce back on the saddle point/critical point and flow to the ergodic fixed point. Look at [C] to see what I mean.

The question now is how to distinguish the finite-size flow. This is the problem which one needs to put an information theoretic/resource bound on.

A general comment on the references. The authors have been very generous with citations (a 7 pages manuscript with more than 120 references is hard to come by). They however managed to avoid mentioning all the works by Boris Altshuler and collaborators, including the paper which started the whole MBL field. I hope this was not intentional, and was just poor knowledge of the literature on the topic of MBL.

Based on the above facts, I cannot suggest the publication of the paper in the present form.

A considerably revised version of the paper can be reconsidered for publication.

Other notes:

Eq. (3) and (A6) make no sense. The authors might want to use the O(\delta) notation like in the rest of the paper. I think the discussion in Append A should really be, in a shorter version, in the main text.

Bibliography

[A] Universality in Anderson localization on random graphs with varying connectivity
P Sierant, M Lewenstein, A Scardicchio
SciPost Physics 15 (2), 045

[C] Renormalization group analysis of the Anderson model on random regular graphs
C Vanoni, BL Altshuler, VE Kravtsov, A Scardicchio
Proceedings of the National Academy of Sciences 121 (29), e2401955121

[D] Many-body localization in the age of classical computing
P Sierant, M Lewenstein, A Scardicchio, L Vidmar, J Zakrzewski
Reports on Progress in Physics 88 (2), 026502

Requested changes

Apply their reasoning beyond the known, finite-dimensional Anderson transition to obtain some new results on infinite-dimensional, Bethe lattice/Regular Random Graph or MBL.

Recommendation

Ask for major revision

---

## Round 2 · Author Response

Dear Editor, Please find attached a revised version of our manuscript.

Critical exponents highlight the universality of phase transitions. Their study allows one to identify the similarities between seemingly-different physical systems and to study them using common tools, thus leading to numerous theoretical discoveries and their experimental confirmation. Information theory, when applied to many-body physics, offers powerful tools for studying universal phenomena. Examining many-body systems in terms of their information content may reveal fundamental limits on many-body physical effects. In our work, we develop an information-theoretical approach to bound critical exponents in disorder-driven phase transitions, relevant to a wide range of physical models, from localization transitions to phase transitions in random quantum circuits. Our approach is intuitive and can be easily applied by other researchers in future studies of other systems.

Beyond developing the method, we apply it to existing models and establish new bounds on the critical exponents in well-studied phase transitions, including various localization transitions and measurement-induced phase transitions in random quantum circuits. Some of our bounds challenge previous approximate analytical or numerical results, allowing us to establish their inadequacy, without the need to identify the relevant mechanisms. This is similar in a sense to using the laws of thermodynamics to rigorously show that a proposed perpetuum mobile cannot work, without the need to identify the exact pitfall of a particular proposal. We would like to thank both Referees for carefully reading our manuscript. We were happy to hear that the first Referee thought that our work is “surpasses expectations and criteria for this Journal; among top 10%”, and we hope that the changes made, along with the discussion below, will be sufficient for the second Referee to support its publication as well. Below we address the comments and suggestions made by the Referees (we quote the Referees’ comments before our response), and describe the changes made to the manuscript. Please also note that following private discussions with colleagues, we have also made more precise the conditions under which one may map random circuits with measurements into percolation in the discussions below Eq. (23) and Eq. (24).

Sincerely yours, Noa Feldman, Niv Davidson, and Moshe Goldstein

Response to comments by the first Referee: 1. “The authors also discuss the case of measurement-induced phase transitions (MIPTs) in random circuits, employing arguments based on entanglement Rényi entropies. I would encourage the authors to extend their analysis to the case of Hilbert space delocalization (see https://doi.org/10.1103/PhysRevLett.128.130605), where significantly larger system sizes are accessible for Clifford circuits. This setting, distinct from MBL, features a transition driven by the subleading coefficient cq rather than the (multi)fractal dimension Dq. Including this analysis would further strengthen the authors’ results and align the presentation of their findings more cohesively.”

We thank the referee for this proposal. In fact, our method does not require further analysis, as the optimal system size in our argument remains unchanged and only the observable allowing one to identify the phase is altered. Examining this additional perspective and confirming that our bound holds further reinforces our result. We have added a reference to this below Eq. (24).

  1. “A noteworthy figure of merit not addressed in the manuscript is that the argument extends to noisy (and open) dynamical systems (see https://doi.org/10.1103/PhysRevX.11.031066, https://doi.org/10.1103/PhysRevLett.132.140401, https://doi.org/10.1103/PRXQuantum.5.030327). In such cases, where z→∞, the bound on remains valid.”

Once again, we thank the referee for this proposal. It allows us to emphasize that the core of our paper is the method or line of thought, which can be applied to a broad range of models. In response, we have added Sec. 4B, referencing the works suggested by the referee, and pointing out some possible new future directions. As explained there, while the random unitaries contain a number of parameters which is exponential in system size, which may lead one to assume that z, in fact the random variables that control the phase transition correspond to the error, which has a constant depth in the temporal direction, and thus corresponds to vanishing z, making this transition a nontrivial test of our theory.

Response to the second referee: 1. “The paper revisits the well-known Harris bound using an information theoretic treatment. The argument is not that different from the usual one: one must have a sufficiently large system/sample size to distinguish two probability distributions which are close to each other. The differentiation is interpreted in terms of "calls" of a classical computer which runs a dynamics. Although the dynamics is quantum, so one should really talk about a quantum computer.”

We emphasize that our argument does not require a quantum computer but rather a classical one that could in principle simulate either a classical or quantum system. As we explain in the manuscript, our argument does not refer to the runtime, which would generally be exponential for simulating a many-body system on a classical computer. Rather we concentrate on the number of random samples (e.g., random potential values) it would require, and comparing them with the inequality in Eq. (3), which originates from classical information theory. The method is also readily applicable to classical models, such as classical percolation, though we primarily focus on quantum models simply because they yield particularly interesting results. While we intended to clarify that our framework relies on classical computation and classical information theory throughout the text, we now recognize that this point may not have been sufficiently explicit. To reinforce this, we have revised the text by changing ‘information-theoretic’ to ‘classical information-theoretic’ in both the introduction and the warm-up example in Sec. 2A.

  1. “Nonetheless, the authors discuss the Anderson model and the "Fock space localization," which is another name for the MBL phase and transition.” Following the referee’s remark, the notation was changed to “MBL in Fock space”. “Here they find a discrepancy between their treatment and the numerics. This is also well known (at least since their ref. [87] was published). They suspect this is due to the fact that the MBL transition is not a vanilla second order phase transition. They might be right (this was also stated in many previous papers, see the discussion in [D]) but they fail to prove their statement.”

As the referee notes, phenomenological evidence and nonrigorous analytical arguments have previously suggested that the many-body localization transition in Fock space does not follow the standard second-order transition paradigm and/or that the scaling exponents found numerically in our Refs. [90,76] are not accurate. However, to the best of our knowledge, our argument provides the first rigorous proof of this fact. The generality of our method ensures the robustness of this result. Naturally, the trade-off for this rigorousness is that we cannot make solid statements beyond the bound itself (as is often the case when deriving rigorous bounds), meaning while we are able to rigorously show that the assumptions and the numerical results of Refs. [90,76] cannot all be true, we are unable to rigorously prove our interpretation of the observed discrepancy.

  1. “One way to do it would be to study the one case in which the Anderson transition itself is not a simple second order phase transitions: this is the limit d\to\infty where it becomes Kosterlitz-Thouless like (see Refs. [A,C]). The authors do not seem to be aware of these papers which instead are central to understand the difference between localization in finite and infinite dimensions, while they cite other papers which have made similar claims but do not present a clear picture of the why a KT transition is present here. If, instead of looking at the leading order term in S_q, like they do in their eq. (11), they looked at the derivative with respect to the logarithm of the system size, they would turn the problem into distinguishing different orbits of the fractal dimensions D_q which evolve as a function of \ln N. Some orbits end on the localized region, some will bounce back on the saddle point/critical point and flow to the ergodic fixed point. Look at [C] to see what I mean. The question now is how to distinguish the finite-size flow. This is the problem which one needs to put an information theoretic/resource bound on.” Following the above, the referee adds that in order to be published, they expect that we “Apply their reasoning beyond the known, finite-dimensional Anderson transition to obtain some new results on infinite-dimensional, Bethe lattice/Regular Random Graph or MBL.”

First let us note that our approach may be applied to the MBL transition in real space to derive the bound 2/d, which was suggested by previous work to hold in this case following the approach by Chayes et al. (and mentioned in Sec. 3A). If the transition is a Berezinski-Kosterlitz-Thouless phase transition, this bound is obeyed since . Since this bound was obtained previously, we did not emphasize it. We have now stated it more explicitly in the beginning of Sec. 3A. For the Fock space MBL transition we also show that q2/d, which could well be obeyed if this transition also displays Berezinski-Kosterlitz-Thouless scaling, leading to q. As for the case of Bethe lattices or regular random graphs, unfortunately, the method in its current form fails to yield a nontrivial bound, due to the exponential dependence of the system size (and in turn, the number of random potential samples N) on the correlation length. It would rather require a nontrivial extension for studying the critical behavior of the correlation depth of the graph, perhaps based on Ref. [96]. We added a paragraph pointing this as a possible future direction at the end of Sec. 3A. We would like to note that the result we obtain for MBL in Fock space is just one out of many new results in our paper. To this result we add the first rigorous bound on dynamical critical exponents in Weyl-like systems (Sec. 2C); bounding the critical exponent of the localization transition with Coulomb interaction, pointing at the inconsistency in Ref. [57] (Sec 3B); obtaining the first analytical result for a general entanglement measure and local Hilbert space dimension for random unitary circuits with measurements (Sec. 4A); and bounding the critical exponent of the error-resilience phase transition, as suggested by Referee 1 (Sec. 4B). We believe that additional results could emerge from this line of thought, as we propose in the future outlook section. Considering all of the above, we believe the paper merits publication in SciPost Phys.

  1. “A general comment on the references. The authors have been very generous with citations (a 7 pages manuscript with more than 120 references is hard to come by). They however managed to avoid mentioning all the works by Boris Altshuler and collaborators, including the paper which started the whole MBL field. I hope this was not intentional, and was just poor knowledge of the literature on the topic of MBL.” Indeed, this is an honest mistake and we thank the referee for pointing it out. The seminal works which brought up the concept of MBL are now cited. “Eq. (3) and (A6) make no sense. The authors might want to use the O(\delta) notation like in the rest of the paper.”

The equations were changed to Theta notation.

  1. “ I think the discussion in Append A should really be, in a shorter version, in the main text.”

A shorter, less formal version of Appendix A appears in Sec. 2A. Following the referee’s remarks, we moved most of the appendix into the main text and left only the formal statement of the protocol (Table 1) in an algorithm form in Appendix A. In addition, we included more details to Fig. 1, which provides a more accessible version of Table 1 from Appendix A.

---

## Round 2 · List of Changes

• Ref. [85] is mentioned below Eq. (24), as a model consistent with our bound.
  • Sec. 4B was added, where we apply our bound noisy quantum circuits with z=0, and discuss previous results on such error models.
  • ‘information-theoretic’ was changed to ‘classical information-theoretic’ in both the introduction and the warm-up example in Sec. 2A.
  • “Fock space localization” was changed to “MBL in Fock space” throughout the text.
  • A paragraph discussing a possible future direction of generalizing our method to random regular graphs or Bethe lattices was added at the end of Sec. 3A.
  • Citations (Refs.21-22, 78, 80, 104, 106, ) were added.
  • Eq. (3) was changed to Theta notation.
  • Most of Appendix A was moved to the main text (Sec. 2A).
  • The conditions under which percolation analysis may be applied in measurement-induced phase transition have been made more precise at the bottom of Sec. IV A.

---

## Editorial Decision

in_refereeing